# The Role of Back Optic Zone Diameter in Myopia Control with Orthokeratology Lenses

**DOI:** 10.3390/jcm10020336

**Published:** 2021-01-18

**Authors:** Jaume Pauné, Silvia Fonts, Lina Rodríguez, Antonio Queirós

**Affiliations:** 1Centre Marsden de Terapia Visual, Consulta 156, Centro Medico Teknon, Vilana 12, 08022 Barcelona, Spain; sifopo92@gmail.com; 2Intitute Visual Clinic Center, Pereira 660002, Colombia; linamar13@yahoo.com; 3Clinical and Experimental Optometry Research Lab (CEORLab), Center of Physics, School of Science, University of Minho, Gualtar, 4710-057 Braga, Portugal; aqp@fisica.uminho.pt

**Keywords:** orthokeratology, myopia progression, optic zone diameter, pupillary diameter, axial length

## Abstract

We compared the efficacy of controlling the annual increase in axial length (AL) in myopic Caucasian children based on two parameters: the back optic zone diameter (BOZD) of the orthokeratology (OK) lens and plus power ring diameter (PPRD) or mid-peripheral annular ring of corneal steepening. Data from 71 myopic patients (mean age, 13.34 ± 1.38 years; range, 10–15 years; 64% male) corrected with different BOZD OK lenses (DRL, Precilens) were collected retrospectively from a Spanish optometric clinic. The sample was divided into groups with BOZDs above or below 5.00 mm and the induced PPRD above or below 4.5 mm, and the relation to AL and refractive progression at 12 months was analyzed. Three subgroups were analyzed, i.e., plus power ring (PPR) inside, outside, or matching the pupil. The mean baseline myopia was −3.11 ± 1.46 D and the AL 24.65 ± 0.88 mm. Significant (*p* < 0.001) differences were found after 12 months of treatment in the refractive error and AL for the BOZD and PPRD. AL changes in subjects with smaller BOZDs decreased significantly regarding larger diameters (0.09 ± 0.12 and 0.15 ± 0.11 mm, respectively); in subjects with a horizontal sector of PPRD falling inside the pupil, the AL increased less (*p* = 0.035) than matching or outside the pupil groups by 0.04 ± 0.10 mm, 0.10 ± 0.11 mm, and 0.17 ± 0.12 mm, respectively. This means a 76% lesser AL growth or 0.13 mm/year in absolute reduction. OK corneal parameters can be modified by changing the OK lens designs, which affects myopia progression and AL elongation. Smaller BOZD induces a reduced PPRDs that slows AL elongation better than standard OK lenses. Further investigations should elucidate the effect of pupillary diameter, PPRD, and power change on myopia control.

## 1. Introduction

Myopia is the most prevalent refractive error and a leading cause of visual impairment and visual loss worldwide [1]. The global prevalence of myopia is also expected to increase significantly [2]. Uncorrected myopia is the second leading cause of preventable blindness worldwide [3], and, therefore, identifying a way to prevent abnormal axial elongation of the eye in children is essential [4].

Animal studies have shown that the mechanisms of optically guided ocular growth are affected by placement of retinal images across a wide area in front of the retina and not solely the fovea [5]. Considering this strong evidence, a range of potential optical interventions to reduce myopic progression has been tested, including bifocal [6] and special ophthalmic lenses [7], soft multifocal lenses [8,9], and orthokeratology (OK) [10,11,12].

Among these interventions, OK shows promise for retarding myopia progression in children and young patients [13]. OK uses rigid gas-permeable lenses with reverse geometry on the back surface. Overnight wear of OK lenses modifies the corneal epithelium, with the central cornea flattened (treatment zone [TZ]) and an annulus of mid-peripheral steepening [14], as demonstrated by a red ring on topographic maps indicating increased corneal plus power. Hence, altering the retinal image profile causes off-axis images to reduce the hyperopic defocus and help to modify eye elongation. Changes induced by OK were significantly correlated with myopia at baseline, with greater changes occurring in association with higher refractions, due to lens design, resulting in a smaller TZ diameter and a stronger plus power ring (PPR) in the mid-periphery [15] (Figure 1). 

As the interest in OK grows, more studies have investigated the possible factors that prevent myopia progression. Several factors have been associated with slower axial length (AL) growth in children treated with OK lenses, such as baseline age [10], refraction [16], higher-order aberrations (HOAs) [17] including coma-like and spherical aberration (SA) [18], asymmetric optical changes (third-order HOAs) [19], TZ decentration [20], corneal relative peripheral power change [21], and peripheral defocus [22]. Previous studies have reported that when the peripheral corneal power, peripheral defocus, HOAs, or SA underwent greater changes, the chances for slowed AL growth increased. Further, previous investigations demonstrating that higher baseline refractive errors treated with OK reported trends toward less axial elongation. Younger subjects generally have lower myopia and larger changes in AL compared to older subjects [23]. However, the change in the relative peripheral refraction at 30° of eccentricity with current OK lenses with a 6-mm back optical zone diameter (BOZD) tend to follow the ratio 1:1 [24,25], meaning a one diopter change in relative peripheral refraction for one diopter change in PPR, and studies evaluating the relative corneal refractive power (RCRP) have pointed to the need for a minimal change of 4.50 D in the RCRP to achieve a chance for 80% myopia control [26]. Hence, customization of OK lenses to obtain an enhanced peripheral plus zone seems suitable for increasing the treatment efficacy in younger subjects or fast progressors.

The first attempts to customize OK lenses tested a smaller OZD and a steeper peripheral tangent but failed to find a significant difference in the peripheral refraction or corneal topographic profile [27]. Other authors reported that the relative peripheral refractive changes differed minimally among three OK lens designs [28]. Moreover, a comparison of Corneal Refractive Therapy lenses (Paragon, Gilbert, AZ, USA) with a sigmoidal corneal proximity “return zone” and a noncurving (tangent) landing and a five-curve Dreamlens (Procornea, Eerbeek, The Netherlands) showed that the latter created a smaller TZ diameter, but despite this, no difference was seen in the plus power profile surrounding the TZ between the two lenses [29]. A later study that evaluated the effect of the BOZD in OK contact lenses regarding the topographic profile in patients with high myopia found that a smaller diameter optical zone (OZ) (5 mm) in OK lenses produced a smaller treatment area and a larger more powerful mid-peripheral ring, which increased the 4th-order spherical aberration that affected only the contrast sensitivity but without differences in visual acuity (VA) and subjective vision compared with a larger OZ diameter (6 mm) [30]. In a study in which the BOZD was reduced by 0.5 mm, the TZ diameter was reliably reduced without detrimentally affecting the lens centration or refractive effect. Reduction of the TZ diameter did not reach significance in the relative peripheral refraction (*p* = 0.058), although a small sample size and measurement artifacts may have masked an effect [31]. 

A recent study that compared the two previously mentioned designs, one with a sigmoidal reverse curve and another with a narrower and steeper reverse zone, reported that the TZ size in subjects wearing a steeper reverse zone was tiny and associated significantly with slower AL growth, indicating that the spatial distribution of the RCRP rather than the total amount may be more important for stopping myopia progression, and those future lenses may be designed with a smaller central OZ [32] or other specific changes on lens design including the reverse zone. However, the association between AL growth and PPR diameter induced by a customized BOZD has not yet been studied in humans.

The current study evaluated for the first time the AL growth induced by different BOZDs in myopic children. The results of this study will enhance our knowledge about the importance of the OK design on future myopia management.

## 2. Methods

### 2.1. Study Design

This retrospective study was based on data collected from subjects fitted for myopia control with OK lenses at a Centre Marsden private optometric clinic between March 2012 and October 2016. It included 71 schoolchildren treated with different BOZDs (range, 4.7–6.0 mm). An expert fitter (JPF) identified the clinical records of subjects with topographic maps that showed centered treatment (<0.5 mm of decentration from visual axis) and a uniform TZ. Only subjects who had been treated successfully, defined as a low residual refractive error (≤0.50 D) and VA (≥6/6 or higher uncorrected VA), were included. Only the right eye of each patient was included in the statistical analysis. One practitioner (JPF) examined all OK wearers. The baseline parameters were collected once refractive and topographic stabilization was obtained.

The patients were followed for 12 months after baseline treatment stabilization. The Ethics Committee for Clinical Research of Centro Medico Teknon approved the study protocol, which adhered to the tenets of the Declaration of Helsinki.

### 2.2. Subjects

The inclusion criteria were age between 10 and 15 years, spherical refractive error between −0.75 and −6.00 D, cylinder with the rule less than −2.00 D, and distant best-corrected VA (BCVA) exceeding 20/20. Subjects with an underlying ocular disease or binocular disorder were excluded.

To calculate the sample size, we assumed a test power of 0.8 and a significance level of 0.05 (two-tailed). The number of subjects required in each group of BOZD was 34.

### 2.3. OK Lens Characteristics

Participants were fitted with a DRL (double reservoir lens) design (Precilens, Creteil, France) following the manufacturer’s protocol that considered the keratometric topographic values, refraction, and corneal diameter. These lenses include a second tear reservoir formed after the reverse curve by a flattened curve coupled with a steepened curve. This second tear reservoir increases hydrodynamic suction forces, which improves centration and faster epithelial changes. All fittings were optimized until centration and the correct refractive outcomes were achieved. Toric designs with or without toric back optic zone radius were used when necessary to obtain the better treatment as possible DRL lenses are made of a Boston XO (hexafocon A) material with oxygen permeability of 100 ISO/Fatt, refractive index of 1.415, Rockwell R hardness of 112 units, and wetting angle of 49 degrees measured with the captive bubble method.

DRL design includes the possibility of customization on BOZD. Reduction of the BOZD is obtained by increasing the width of peripheral curves to keep the total diameter constant, and no changes were made in reverse curve width. Curvature of the reverse curve was adjusted by the manufacturer so that the fitting of the lens with a given cornea remains unchanged.

### 2.4. Outcomes

The refractive error was measured in 0.01-D increments with cycloplegic autorefraction using the Grand Seiko Autorefractometer/Keratometer WAM-5500 (Grand Seiko Co., Ltd., Hiroshima, Japan) [33] with the OK lens on the eye in all visits once the lens was centered between blinks, which is achieved considering the diameter and fitting characteristics achieved with the DRL lenses. Cycloplegia was achieved using two drops of cyclopentolate hydrochloride 1.0% (Alcon, El Masnou, Spain) instilled 10 min apart. The same examiner performed and averaged six consecutive measurements 30 min after the second drop was instilled. Keratometric readings were performed on the anterior lens surface to adjust the final refraction to any shape changes the lens could have undergone over time by flexure. The AL was measured in 0.01-mm steps under cycloplegia obtained using cyclopentolate hydrochloride 1.0% and anesthesia using oxybuprocaine hydrochloride 0.4% and tetracaine hydrochloride 0.1% (Alcon) using the OcuScan RxP Ophthalmic Ultrasound System [34] (Alcon, Fort Worth, TX, USA). 

Echographic signals were examined for relatively equal lens peaks and well-defined retinal peaks. The same experienced optometrist performed 10 consecutive measurements. When poor signals were detected, the measures were repeated. The mean axial dimensions were calculated as the mean of the 10 readings.

Tangential topographic maps were retrieved first and before cyclopegia using the Keratron Onda [35] (Keratron, Rome, Italy). The points of higher plus power change (steeper curvature radius) in the PPR were identified for baseline and 12-month treatment, and the PPR diameters (PPRDs) for horizontal and vertical meridians were retrieved and the median value was obtained. Pupillary size was obtained directly from the topographic data obtained under ambient mesopic room illumination, but photopic conditions were still considered due to topographer intrinsic light level [36].

To better perceive the annual AL increase in myopic children, the two analysis parameters, i.e., the BOZD and PPRD, were divided into two groups (larger and smaller diameters). Thirty-six subjects comprised the group with a BOZD exceeding 5 mm (L-BOZD) and 35 with a BOZD equal to or smaller than 5 mm (S-BOZD). For the plus power ring diameter (PPRD) with larger PPRDs when they were >4.5 mm (L-PPRD, *n* = 36) and smaller PPRDs when they were ≤4.5 mm (S-PPRD, *n* = 35).

To evaluate the relation of the pupillary diameter with AL growth (considering that the widths of PPR fall between 2.4 and 1.9 mm [31]), a 0.9-mm distance from the center of the PPR (80% of the mean value) was adopted. Based on these assumptions, three groups were created: no effect (NE, *n* = 23) of the PPRD on the pupil (a pupillary diameter <PPRD−0.9 mm), medium effect (ME, *n* = 40) of the PPRD on the pupil (the pupil settled into the span of the PPRD ± 0.9 mm), and full effect (FE, *n* = 8) of the PPRD on the pupil when the pupil exceeded PPRD + 0.9 mm.

### 2.5. Statistical Analysis

Statistical analyses were performed using SPSS software version 25.0 (SPSS Inc., Chicago, IL, USA). The Kolmogorov–Smirnov test was applied to assess the normality of the data distribution. The paired sample *t*-test or Wilcoxon signed-ranks test was used for comparisons between two different conditions for normally or non-normally distributed variables, respectively. An independent *t*-test or Mann-Whitney test was used to compare the means of continuous variables between different groups, whereas a paired *t*-test was used to compare the changes in the measurement results for paired samples. The chi-square test or Fisher’s exact test was used to examine differences with categorical variables. Stepwise logistic regression analysis was performed on the dataset to determine the best predictors of faster AL growth (≥0.10 mm/year). The ALs in individuals who are not myopic can increase up to 0.10 mm/year on average without developing myopia [23]. For statistical purposes, *p* < 0.05 was considered significant. 

## 3. Results

The ages during which the patients (41 boys, 30 girls) wore OK lenses ranged from 10 to 15 years (mean ±  standard error, 13.34 ± 1.38 years). At baseline, the myopia ranged from −0.75 to −6.00 D (mean, −3.11 ± 1.46 D), and the astigmatism ranged from 0.00 to −2.00 D (mean −0.59 ± 0.40 D). The baseline demographic and ocular characteristics are shown in Table 1; no significant differences were seen between the OK groups. The samples in the baseline showed a normal distribution. No differences were found between genders.

### 3.1. BOZD

The overall trends in the changes in the refractive error (mean), vitreous chamber, and AL per 12-month period in the OK groups are shown in Table 2, according to the BOZD diameter.

The L-BOZD group had a PPRD of 5.05 ± 0.47 mm and S-BOZD 4.12 ± 0.32 mm (*p* < 0.001). Analysis of the difference in the refractive error at 12 months (final-baseline) showed significant differences between the L-BOZD group with a mean myopic increment of −0.27 ± 0.23 D and the S-BOZD group with a mean myopic decrease of +0.16 ± 0.34 D (*p* < 0.001, independent *t*-test). The AL increased significantly less in the S-BOZD group compared with the L-BOZD group (*p* = 0.007, independent *t*-test), i.e., 0.08 ± 0.12 and 0.16 ± 0.11 mm, respectively. The distribution between groups showed that the AL in most S-BOZD subjects increased less than 0.10 mm/year, and the opposite trend was seen in the L-BOZD group with the AL increasing by >0.10 mm/year (Figure 2).

### 3.2. PPRD

No differences between groups were found in age, sex, keratometry, or eccentricity. Horizontal PPRDs were slightly larger than vertical PPRDs, i.e., 4.68 ± 0.60 and 4.50 ± 0.64, respectively (*p* < 0.001, independent *t*-test). L-PPRDs were characterized by smaller pupillary diameters than the S-PPRDs, i.e., 4.01 ± 0.53 mm and 4.45 ± 0.68 mm, respectively (*p* < 0.004, independent *t*-test).

The S-PPRD refractive M value decreased after 1 year (mean, 0.12 ± 0.35 D), and the M L-PPRD increased (mean, −0.23 ± 0.28 D) (*p* < 0.001). Slower AL growth was found in the S-PPRD group compared with the L-PPRD group, i.e., 0.09 ± 0.12 mm and 0.15 ± 0.11 mm, respectively (*p* = 0.030 independent *t*-test). The anterior chamber, lens, or vitreous chamber did not differ significantly between the groups. The correlation between the BOZD and PPRD was seen (Figure 3) (r = 0.827; PPRD=0.77×BOZD+0.35).

For the mean PPRD, we found significant differences only between the NE and FE groups (*p* = 0.028, independent *t*-test), but no other combinations were significant. Horizontal PPRD was significantly different between groups (*p* = 0.035, ANOVA). The data showed large increases in the AL in the NE group, medium increases in the ME group, and small increases in the FE group; AL_NE_ = 0.17 ± 0.12 mm, AL_ME_ = 0.10 ± 0.11 mm, and AL_FE_ = 0.04 ± 0.010 mm, respectively. The vertical PPRD even following a similar tendency was not significant (*p* = 0.1589, ANOVA), and the mean PPRD was close to statistical significance (*p* = 0.056, ANOVA). The change in myopia was greater in the NE group than in the ME group and higher than in the FE group where we found positive shifts, i.e., −0.25 ± 0.31 D, −0.01 ± 0.29 D, and +0.27 ± 0.50 D, respectively. Results are shown in Table 3 and plotted in Figure 4.

Multivariable correlation analyses showed that the factor associated significantly with control of AL progression of 0.10/year was the BOZD (x^2^(1) = 5.326; *p* = 0.021, r^2^_Nagelkerke_ = 0.097). Age, myopia, pupillary diameter at baseline, and PPRD did not reach significance. We found a significant correlation between the BOZD and AL growth (odds ratio, 2.406; confidence interval, 1.111–5.212) that provided the following formula: $$\Delta AL=\frac{e^{-4.564+0.878\times \mathrm{BOZD}}}{1+e^{-4.564+0.878\times \mathrm{BOZD}}}$$

## 4. Discussion

This study showed that a change in the BOZD in the same OK lens design, without other major changes, modifies the TZ diameter, especially the diameter of the steepened mid-peripheral corneal annulus, modifying almost certainly the relative peripheral myopic shift and increasing HOAs (including SA and Coma) and associated optical factors related to reduced myopia progression in children.

The children in our study using OK lenses with a smaller BOZD showed a reduced PPRD (*p*< 0.001), and the group with a PPRD smaller than 4.5 mm showed a hyperopic shift in refraction and slower AL growth after one year. In a previous article and in a sample of Caucasian Spanish children with the same characteristics of age, sex distribution, and refractive error at baseline [9,37], we reported 0.15 ± 0.10 mm AL growth at 12 months with the 6.0-mm BOZD, which agrees with the current results. AL growth seen in the 12-month data for the control group in which single-vision glasses were worn was 0.28 ± 0.17, indicating a 0.13 mm/year AL difference in absolute value and a control effect of 46%, which agreed with the overall consensus [38]. Our values also were correlated with another study of younger Caucasian Spanish children (mean age, 10 years) who had 0.38 mm/year AL growth in the control group, which was 0.15 mm more than in the OK group. Regarding the current results in children with a PPRD smaller than 4.5 mm, AL growth of 0.09 ± 0.12 mm was seen, which accounts for 0.19 mm/year less AL growth in cumulative absolute reduction in axial elongation (CARE) [39] and a control effect of 68%.

Chen et al. reported greater myopia control in those with larger-than-average pupils and a negative effect (myopia progressed faster than the controls) in those with smaller-than-average pupils [40]. The authors suggested that the larger pupils allowed more of the corneal plus powered change that forms around the OK-induced TZ to fall inside the pupil. This, in turn, resulted in a larger area of the peripheral retina with myopia to defocus and consequently caused greater myopia control [22]. This hypothesis seems to agree with the myopic shift seen in the visual Strehl ratio based on the modulation transfer function peripheral refraction profile when the pupillary diameter increased from 3 to 6 mm in a study using ray-tracing software for different pupillary sizes [41]. Experimental studies in primates supported the concept that the optical signals that have a decisive role in regulating ocular growth and refractive development [42] seem to be dose-dependent [43]. Moreover, a significant relationship between ocular HOAs and axial growth indicated that greater levels of higher-order root mean square slow axial elongation [7,44]. 

Pupil size, which is considered a very dynamic parameter, is, as part of the near vision triad (accommodation, convergence, and miosis), influenced by working distance, as well as by the level of illumination under which each task is conducted. Furthermore, even within the same task, illumination, and working distance, pupil diameter has been shown to present with significant differences between individuals [45]. A study to determine pupil diameter under different real situations found statistically significant differences between in-office and daily-life conditions [46]. These discrepancies may be assumed to lead to relevant differences in the light distribution, thus affecting the optical pathways related to myopia control. Pupillary size is affected by different measurement devices where visible light is utilized [47]. Nonetheless, regarding illumination, it may be safely assumed that tasks presumably related to myopia are undertaken under photopic conditions similar to those under which the topographic pupillary diameter was assessed. Although no standardized method has been described in the literature for measuring the pupillary size regarding myopia control, this is the most frequently used clinical method to measure the pupillary diameter by professionals. Furthermore, it had been found to be highly repeatable and suitable for use on children [48]. When the study sample was divided into three subgroups regarding the effect of the mid-peripheral plus power annulus in the pupil, there seemed to be a dose–response relationship within the groups. Accordingly, the group that was considered to be unaffected by the PPR had a greater increase in AL. However, in the group in which the total surface of the PPR was inside the pupil area, reduced AL growth was seen, meaning that the FE group has a 77% shorter AL after 12 months compared with the NE group, 0.04 ± 0.04 versus 0.17 ± 0.02, respectively, which accounts for 0.13 mm in absolute value. Moreover, the horizontal PPRD reached significance (*p* = 0.035) but not vertical or mean PPRD regarding AL; regarding refractive M outcomes, the differences were significant (*p* < 0.001), with an absolute difference of 0.52 D/year, i.e., FE +0.27 ± 0.50 D and NE −0.25 ± 0.31 D.

Therefore, we suggest for the first time a relationship between pupillary size and PPR to improve the ability to slow myopia in children. With normal AL growth for emmetropic children considered a change of 0.09 mm/year at 12 years of age [49], these results mean that refractive change may be halted or reduced in many children with a change in lens geometry. Since our results involve important implications for OK lens design in the future, this area needs further investigation. However, this study presents ages above 10 years (mean age 13.34 ± 1.38 years) and, as reported by Queirós et al. [23], the effect of orthokeratology on axial lengthening >0.10 mm/year is much more effective above 11 years, so these results must be looked at cautiously, as the findings may not apply to younger ages when axial length in increasing more rapidly. A limitation of the current study was the small sample sizes as the subjects were subdivided into the three groups; the fact that in our sample, the children tended to have small photopic pupillary sizes reduced the number of children in whom the entire PPR width remained inside the pupil area, resulting in a small sample and statistical weakness. Future studies with larger samples should elucidate the importance of the relationship between pupillary diameter and PPR width and power. The photopic pupillary size was selected to mimic real-world situations and near tasks. Notwithstanding, the average differences in pupillary diameters between scotopic and photopic conditions remain largely constant (1.5 mm) across the range of ages from 18 to 62 years [50]. Future studies should confirm the current results.

Even though all subjects in the current study had TZ decentration lower than 0.5 mm, this is a common phenomenon in OK that mostly happens toward the inferotemporal corneal quadrant [51], and it has been associated with reduced AL growth [20,52]. Thus, even if the pupillary diameter is smaller than the PPR edges, there is likely to be a quadrant effect on the retinal image. Indeed, the temporal retina is affected by changes in the nasal cornea that tend to be closer to the apex due to lens decentration. We are unaware of how this may have affected our data.

A possible limitation of the study was the less precise measurement of axial elongation with A-scan ultrasonography compared to partial coherence interferometry. However, ultrasound biometry has been largely used in longitudinal studies of myopia in children as in the CLEERE [53] and COMET [54] studies. A-scan ultrasonography has shown variability in the overall axial length of 0.06 ± 0.04 mm and is a useful technique to assess changes in ocular components in children [55]. Furthermore, during the study, the measurement methods did not vary, and one expert optometrist performed all measurements. Although we cannot ignore the possibility of an excess of pressure when data were acquired, the changes in Vitreous Chamber Depth also were significant and followed the overall trend. Furthermore, any bias in the biometric measurements will result in narrower Anterior Chamber Depth and shorter AL measures. This error in systematic measurement acquisition will not modify the results.

Another limitation was that the patients were not randomly assigned to different treatment groups. Regarding this, we included covariates such as baseline AL, age, and gender to the linear mixed-effect model to adjust for observed differences between the two groups. Despite these efforts, we believe that a randomized clinical trial is preferable to control for other unmeasurable variants between the two groups.

A significant number of subjects had decreased AL values. Previous studies [56] also reported reduced ALs. The axial shortening in the present study can be attributed to forward retinal movement, presumably related to choroidal thickening that was previously reported during OK [57]. However, this cannot be confirmed because the choroidal thickness was not measured.

## 5. Conclusions

Altering the OK lens design can reliably modify the annular PPRD. The current study provided evidence that a smaller BOZD with DRL OK design reduces the PPRD and improves the effect of OK to slow axial growth in myopia by displacing the steepened annular ring in OK closer to the central zone interacting with the pupil. We show a significant difference in AL by 0.13 mm/year and 0.52 D/year in M value when the horizontal section of the PPR fell inside or outside the pupillary diameter, which accounts for a 77% shorter AL after 12 months. The difference showed a dose response of the system. Longitudinal studies are needed to assess if a smaller BOZD OK lens design increases the efficacy in reducing AL growth during OK and the pupillary role.

## Figures and Tables

**Figure 1 jcm-10-00336-f001:**
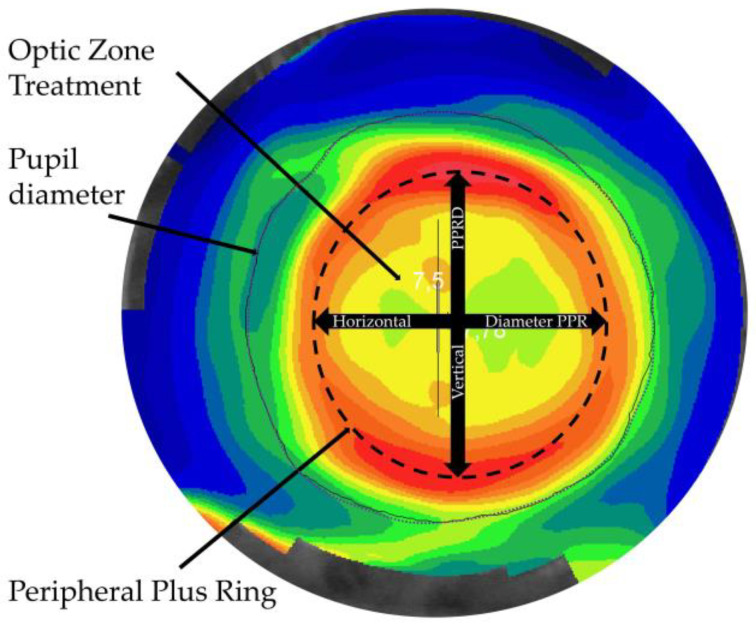
A tangential topographic map shows the plus power ring (PPR) diameter analyzed in this study.

**Figure 2 jcm-10-00336-f002:**
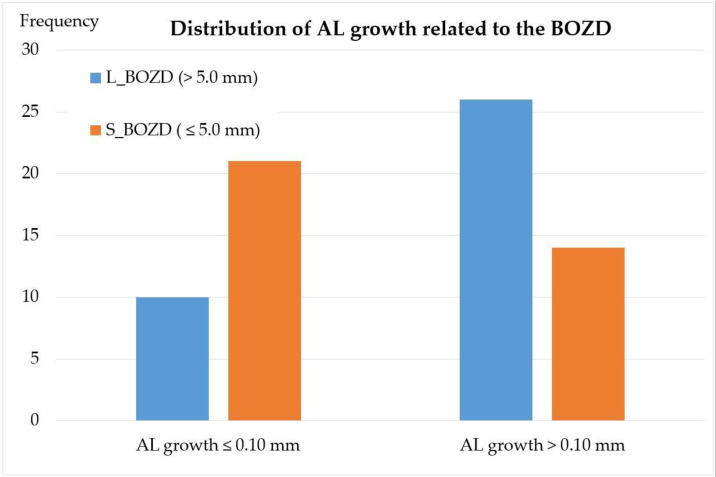
Distribution of subjects with more than 0.10 mm or less than 0.10 mm of AL change annually in relation to BOZDs higher or lower than 5.0 mm in diameter. A trend for more subjects with slower AL growth is seen in association with reduced BOZD. AL: axial lenght, L_BOZD: large back optic zone diameter, S_BOZD: small back optic zone diameter.

**Figure 3 jcm-10-00336-f003:**
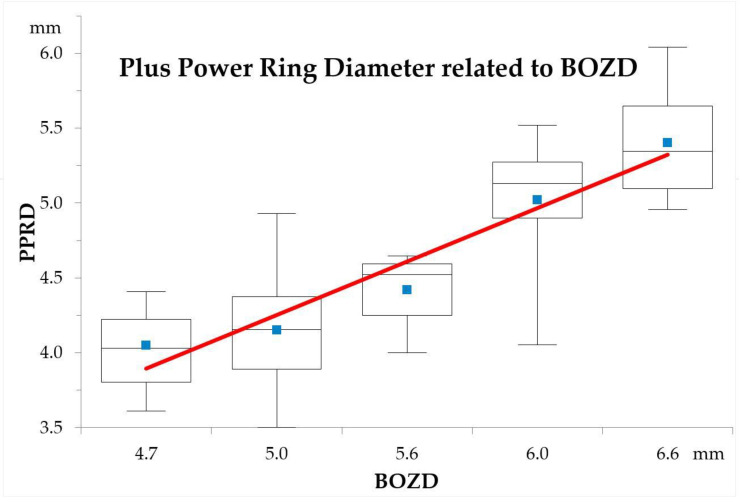
The correlation between the back optic zone diameter (BOZD) of the lenses and the plus power ring diameter (PPRD). The different PPRD sizes used in the study are represented by the mean values (blue squares), median values (flat lines), 1st and 3rd quartile interval boxes, and minimal and maximal values. The polynomial line shows the correlation between groups.

**Figure 4 jcm-10-00336-f004:**
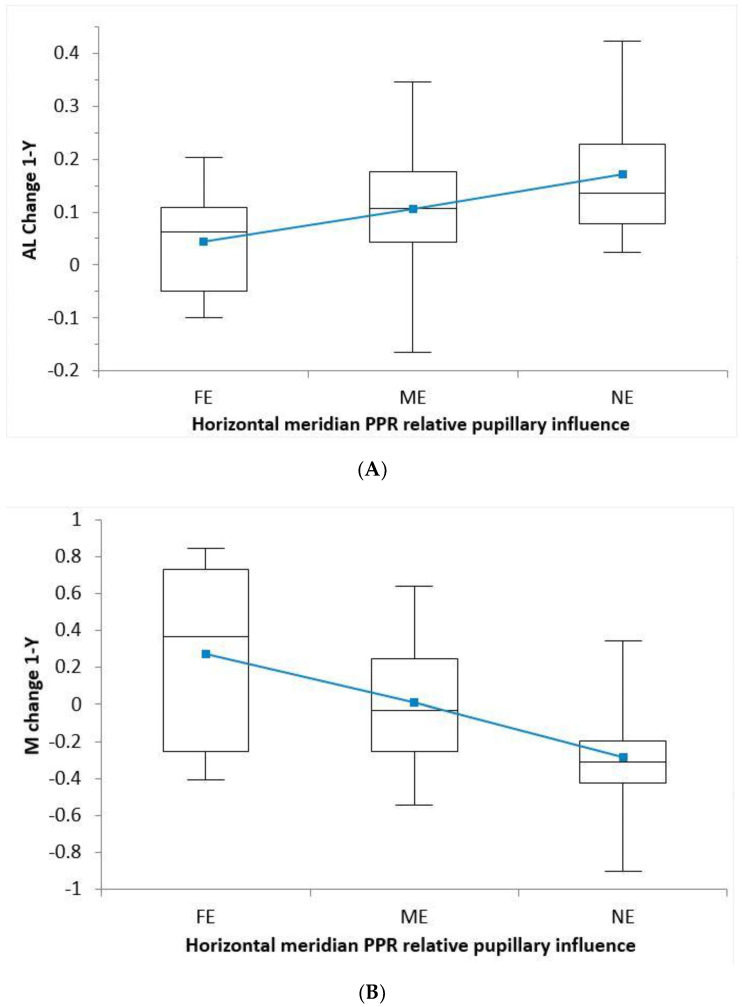
(**A**) The relative plus power ring diameter (PPRD)/pupillary effect for horizontal meridian to axial length (AL) at one year change. Vertical meridian PPRD and mean PPRD resulted in a very similar image. The group with a full effect (FE) had a pupillary diameter larger than the PPRD + 0.9 mm, the group with medium effect (ME) had a pupil between the PPRD ± 0.9, and the No Effect (NE) group had a pupillary diameter smaller than the PPRD − 0.9 mm. (**B**) The relative plus power ring diameter (PPRD)/pupillary effect for horizontal meridian related to spherical refraction (M) at 12 month increase. Vertical meridian PPRD and mean PPRD resulted in a very similar image and was not plotted. In figure A and B, a trend toward a dose-dependent PPRD is observed. The mean values are represented by small squares (blue); medians are shown by flat lines, 1st and 3rd quartile interval boxes; and the minimal and maximal values are plotted.

**Table 1 jcm-10-00336-t001:** Summary of the baseline demographic data and refractive errors of the study subjects by BOZD group (>5.0 and ≤5.0 mm).

Baseline	BZOD Group	Mean ± SD	Minimum	Maximum	*p*-Value
Age (years)	>5	13.27 ± 1.50	10.08	15.31	0.666 *
	≤5	13.41 ± 1.25	11.00	15.82	
Flat keratometry (mm)	>5	7.87 ± 0.27	7.47	8.45	0.580 *
	≤5	7.92 ± 0.37	7.02	8.48	
Steep keratometry (mm)	>5	7.72 ± 0.27	7.21	8.21	0.410 *
	≤5	7.78 ± 0.34	6.96	8.44	
Eccentricity flat	>5	0.43 ± 0.07	0.22	0.55	0.845 **^†^**
	≤5	0.43 ± 0.08	0.28	0.55	
Pupillary diameter (mm)	>5	4.01 ± 0.50	3.35	5.39	0.004 *
	≤5	4.45 ± 0.71	3.22	6.28	
BOZD (mm)	>5	6.11 ± 0.34	5.60	6.60	0.000 **^†^**
	≤5	4.91 ± 0.14	4.70	5.00	
PPRD Horizontal	>5	5.13 ± 0.46	4.03	6.08	0.000 *
	≤5	4.21 ± 0.30	3.73	4.88	
PPRD Vertical	>5	4.96 ± 0.49	3.97	6.00	0.000 *
	≤5	4.02 ± 0.39	3.17	4.98	
Mean PPRD (mm)	>5	5.05 ± 0.47	4.00	6.04	0.000 *
	≤5	4.12 ± 0.32	3.50	4.93	
Axial Length (mm)	>5	24.68 ± 0.94	23.14	26.37	0.789 *
	≤5	24.61 ± 0.83	23.16	26.14	

BOZD: back optical zone diameter; PPRD: plus power ring diameter; SD: standard deviation. * independent *t*-test; **^†^** Mann-Whitney Test.

**Table 2 jcm-10-00336-t002:** Changes from baseline to 12-months treatment in the BOZD group: >5 mm (L-BOZD, *n* = 36) and ≤5 mm (S-BOZD, *n* = 35).

	BOZD (mm)	Baseline	12 Months	12-Month-Baseline	*p* Value
Spherical equivalent (Diopters)	>5	−3.41 ± 1.51	−3.68 ± 1.51	−0.27 ± 0.23	<0.001 *
	≤5	−2.80 ± 1.37	−2.64 ± 1.45	0.16 ± 0.34	0.013 **^†^**
	*p*	0.079 **^‡^**	0.003 **^§^**	<0.001 **^‡^**	
Vitreous chamber depth (mm)	>5	17.31 ± 0.95	17.40 ± 0.99	0.09 ± 0.12	<0.001 *
	≤5	17.16 ± 0.77	17.21 ± 0.77	0.05 ± 0.12	0.012 **^†^**
	*p*	0.468 **^‡^**	0.359 **^‡^**	0.311 **^§^**	
Axial Length (mm)	>5	24.69 ± 0.94	24.84 ± 0.96	0.16 ± 0.11	<0.001 *
	≤5	24.61 ± 0.83	24.69 ± 0.85	0.08 ± 0.12	<0.001 *
	*p*	0.723 **^‡^**	0.488 **^‡^**	0.007 **^‡^**	

BOZD: back optical zone diameter; * Paired sample *t*-test; **^†^** Wilcoxon test, **^‡^** independent *t*-test; **^§^** Mann–Whitney test.

**Table 3 jcm-10-00336-t003:** Changes of AL and refractive outcome (M) from baseline to 12-months treatment in the PPRD group: no effect (NE, *n* = 23) of the PPRD on the pupil (a pupillary diameter <PPRD−0.9 mm), medium effect (ME, *n* = 40) of the PPRD on the pupil (the pupil settled into the span of the PPRD ± 0.9 mm), and full effect (FE, *n* = 8) of the PPRD on the pupil when the pupil exceeded PPRD + 0.9 mm.

	PPRD	FE	ME	NE	*p*
AL	Horizontal	0.04 ± 0.10	0.10 ± 0.11	0.17 ± 0.12	0.035 *
	Vertical	0.06 ± 0.11	0.11 ± 0.11	0.16 ± 0.12	0.158 *
	Mean	0.04 ± 0.10	0.11 ± 0.11	0.17 ± 0.12	0.056 *
M	Horizontal	0.27 ± 0.50	0.01 ± 0.30	−0.29 ± 0.26	<0.001 **^†^**
	Vertical	0.28 ± 0.47	−0.02 ± 0.29	−0.24 ± 0.31	<0.001 **^†^**
	Mean	0.27 ± 0.50	−0.01 ± 0.29	−0.25 ± 0.31	<0.001 **^†^**

PPRD: plus power ring diameter; SD: standard deviation; AL: axial length; M: Spherical equivalent; FE: Full effect; ME: medium effect; NE: no effect.* Kruskall–Wallis; **^†^** ANOVA.

## Data Availability

The data presented in this study are available on request from the corresponding author. The data are not publicly available due toprivacy reasons.

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
