# Peer review of "The Role of Back Optic Zone Diameter in Myopia Control with Orthokeratology Lenses"

_jcm, 2021, doi:10.3390/jcm10020336_

Round 1

Reviewer 1 Report

Review for JCM 12/22/20

First, great study question that needs addressing.

Overall, there are some major issues with the study design that make it hard to validate these findings.

First, it is a retrospective study with no control group. Second, AL measurements are done with ultrasound that has more variability than optical biometry. The fact that there is only one practitioner taking all measurements is better than if other techs or doctors were taking these but it still calls into question the accuracy necessary to make your conclusions based on ultrasound findings. Third, pupil readings were done in photopic conditions in front of the topographer. These may or may not represent the average pupil diameter over the course of a normal day during various activities (indoors, outdoors, and near vs distance). Finally, studies have shown variable patient responses to OrthoK lenses of identical parameters. Ie. The diameter of the TZ created by lenses of identical BOZD may be quite different for different patients. Therefore, there may be more factors in play than just BOZD that determine the resulting size of the TZ as measured on topography.

In the abstract, It may be a technical point but it needs clarification. You state the two measures are of the lens “… on two parameters of the orthokeratology (OK) lens: the back optic zone diameter (BOZD) and plus power ring diameter (PPRD) or mid-peripheral annular ring of corneal steepening.” Actually, the BOZD is a parameter of the lens but the PPRD is a corneal parameter of the post-treatment topography, not the lens.

Line 48, I suggest changing the word “stop” to “slow or minimize” as nothing has been shown to stop axial elongation.

Line 66, re-word the sentence as change in higher refractions does not “result” in smaller TZ diameter. It is not the higher refraction that causes this but rather it is the design changes for higher refraction that may have more effect on TZ diameter.

Line 80, re-word to say these characteristics are “associated” with slower axial elongation

Line 116, how was it associated , positive or negative?

Line 130, other than the difference in BOZD, were there any other differences in the design used for the subjects? Ie. Width of other curves, OAD, determination of curvature of reverse curves.

Line 175, was the over-refraction also with auto-refractor or was it subjective?

Line 399, you can not attribute axial shortening to epithelial changes because epi thickness in OrthoK changes approx. 10 microns (0.01 mm) which is the limit of accuracy of optical biometry and less than the accuracy of ultrasound biometry.

Although mentioned in the discussion, age is a difficult factor to work around in studies on axial elongation. The mean age of subjects in this study is over 13, an age where AL increases are slowing compared to 7 to 10 year-olds. These results must be looked at cautiously as the findings may not apply to younger ages when axial length in increasing more rapidly.

I think you must also acknowledge that there may be other factors that influence the rate of Axial elongation during OrthoK. Peripheral plus is an important factor but there may be many others that have not been adequately studied.

If you have the data, it might be interesting to see if there is any difference in AL elongation between a group of subjects who all have the same pupillary diameter but different PPRD.

Author Response

We appreciate so much your valuable observations. They help us to a better exposition and construction of the article. We attach a word document with the observations and changes made on the document. Please see that attachement.

We remain at your disposal for further observations and suggestions

First, great study question that needs addressing.

Overall, there are some major issues with the study design that make it hard to validate these findings.

In the first place, we appreciated so much all these valuable observations and suggestions. We will try to answer all the questions one by one on the following lines.

Thank you very much for your comment and review.

First, it is a retrospective study with no control group.

The design of the study included a self-control study to avoid much interference factors. Indeed, we generated two groups for BOZD and three groups for PPRD that acts as a control group between them. Moreover, we intended to know the influence on Axial Length growth from the topographical changes, specifically from the diameter of the PPR. Honestly, we are unaware of different characteristics for a control group in a different approach than we already did.

Second, AL measurements are done with ultrasound that has more variability than optical biometry. The fact that there is only one practitioner taking all measurements is better than if other techs or doctors were taking these but it still calls into question the accuracy necessary to make your conclusions based on ultrasound findings.

We acknowledge this and we include in discussion the following paragraph.

A possible limitation of the study was less precise measurement of axial elongation with A-scan ultrasonography compared to partial coherence interferometry. However, ultrasound biometry had been largely used in longitudinal studies of myopia in children as in the CLEERE [Mutti, D.O.;Hayes, J. R.; Mitchell, G. L. et al., Refractive error, axial length, and relative peripheral refractive error before and after the onset of myopia, Investigative Ophthalmology and Visual Science, vol. 48, no. 6, pp. 2510–2519, 2007.]  and COMET [Gwiazda, J.; Hyman, L.; Hussein, M. et al., A randomized clinical trial of progressive addition lenses versus single vision lenses on the progression of myopia in children, Investigative Opthalmology & Visual Science, vol. 44, no. 4, pp. 1492–1500, 2003.] studies. A-scan ultrasonography showed a variability of overall axial length of 0.06 +/- 0.04 mm, and is therefore a useful technique to assess changes in ocular components in children. [Kurtz, D.; Manny, R. and Hussein, M. Variability of the ocular component measurements in children using A-scan ultrasonography, Optometry and Vision Science, vol. 81, no. 1, pp. 35–43, 2004.]. In addition, during the study, the measurement methods did not vary and one expert optometrist performed all measurements. Regardless of the fact that we may not discard an excess of pressure when data was acquired, the changes in VCD also was significant and followed the overall trend. Furthermore, any bias in the biometric measurements will result in narrower ACD and shorter AL measures. This error in systematic measurement acquisition will not modify the results.

Third, pupil readings were done in photopic conditions in front of the topographer. These may or may not represent the average pupil diameter over the course of a normal day during various activities (indoors, outdoors, and near vs distance).

We understand the issue and we add this paragraph into discussion

Considered a very dynamic parameter, pupil size, as part of the near vision triad (accommodation, convergence and miosis), is influenced by working distance, as well as by the level of illumination under which each task is conducted. In addition, even within the same task, illumination and working distance, pupil diameter has been shown to present with significant differences between individuals. [Koch D.D., Samuelson S.W., Haft E.A., Merin L.M. Pupillary size and responsiveness. Implications for selection of a bifocal intraocular lens. Ophthalmology. 1991;98:1030–1035.] A study to determine pupil diameter under different real situations found statistically significant differences between in-office and daily life conditions [Cardona, G., López, S. Pupil diameter, working distance, and illumination during habitual tasks. Implications for simultaneous vision contact lenses for presbyopia. J Optom. 2016 Apr-Jun;9(2):78-84.]. It may be assumed that these discrepancies would lead to relevant differences in the light distribution, thus influencing optic pathways related to myopia control. Pupils size also suffers from different measurement devices where visible light is utilized [Hsieh, YT; Hu, FR. The correlation of pupil size measured by Colvard pupillometer and Orbscan II. J Refract Surg. 2007 Oct;23(8):789-95. PMID: 17985798.]. Nonetheless, in regards to illumination, it may be safely assumed that tasks presumably related to myopia are undertaken under photopic conditions close to which topography pupil diameter was assessed. Furthermore, Pupil size measured with corneal topographers is the most widely extended method under clinical conditions making the results of this study more relevant for the average practitioner.

Finally, studies have shown variable patient responses to OrthoK lenses of identical parameters. Ie. The diameter of the TZ created by lenses of identical BOZD may be quite different for different patients. Therefore, there may be more factors in play than just BOZD that determine the resulting size of the TZ as measured on topography.

As the reviewer suggests the TZ diameter created by an Orthokeratology lens is dependent on the design, the fitting, and corneal factors. Even in the introduction we may read this we added the following:

“A recent study that compared the two previously mentioned designs, one with a sigmoidal reverse curve and another with a narrower and steeper reverse zone, reported that the TZ size in subjects wearing a steeper reverse zone was tiny and associated significantly with AL growth, indicating that the spatial distribution of the RCRP rather than the total amount may be more important for stopping myopia progression and those future lenses should be designed with a smaller central OZ [32]. or other specific changes on lens design as the reverse zone.

And in discussion:

This study showed that a change in the BOZD in the same OK lens design changed the TZ diameter;

In the abstract, It may be a technical point but it needs clarification. You state the two measures are of the lens “… on two parameters of the orthokeratology (OK) lens: the back optic zone diameter (BOZD) and plus power ring diameter (PPRD) or mid-peripheral annular ring of corneal steepening.” Actually, the BOZD is a parameter of the lens but the PPRD is a corneal parameter of the post-treatment topography, not the lens.

We compared the efficacy of controlling the annual increase in axial length (AL) in myopic Caucasian children based on two parameters:  of the orthokeratology (OK) lens: the back optic zone diameter (BOZD) of the orthokeratology (OK) lens and plus power ring diameter (PPRD) or mid-peripheral annular ring of corneal steepening

Line 48, I suggest changing the word “stop” to “slow or minimize” as nothing has been shown to stop axial elongation.

Done

Line 66, re-word the sentence as change in higher refractions does not “result” in smaller TZ diameter. It is not the higher refraction that causes this but rather it is the design changes for higher refraction that may have more effect on TZ diameter.

Changes induced by OK were significantly correlated with myopia at baseline, with greater changes occurring in higher refractions, due to lens design, resulting in a smaller TZ diameter and a stronger plus power ring (PPR) in the mid-periphery [15]

Line 80, re-word to say these characteristics are “associated” with slower axial elongation

“Several factors have been associated with slower axial length (AL) growth in children treated with OK lenses, such as baseline age”

Line 116, how was it associated, positive or negative?

…reported that the TZ size in subjects wearing a steeper reverse zone was tiny and associated significantly with slower AL growth ….

Line 130, other than the difference in BOZD, were there any other differences in the design used for the subjects? Ie. Width of other curves, OAD, determination of curvature of reverse curves.

DRL design includes customization on BOZD. Reduction of BOZD was obtained increasing the width of peripheral curves to keep overall diameter constant, and no changes in reverse curve width was made. Curvature of reverse curve was adjusted to keep sagittal depth constant.

Line 175, was the over-refraction also with auto-refractor or was it subjective?

autorefractometer

Line 399, you cannot attribute axial shortening to epithelial changes because epi thickness in OrthoK changes approx. 10 microns (0.01 mm) which is the limit of accuracy of optical biometry and less than the accuracy of ultrasound biometry.

We agree with the reviewer. This part has been deleted

Although mentioned in the discussion, age is a difficult factor to work around in studies on axial elongation. The mean age of subjects in this study is over 13, an age where AL increases are slowing compared to 7 to 10 year-olds. These results must be looked at cautiously as the findings may not apply to younger ages when axial length in increasing more rapidly.

We agree with the reviewer. This observation was added in the discussion

However, this study presents ages above 10 years (mean age 13.34±1.38 years) and as reported by Queirós et al. [ Queirós, A. et al. Refractive, biometric and corneal topographic parameter changes during 12 months of orthokeratology. Clin. Exp. Optom. 2020, 103, 454–462.] the effect of orthokeratology on axial lengthening >0.10mm/year is much more effective above 11 years, so these results must be looked at cautiously as the findings may not apply to younger ages when axial length in increasing more rapidly.

I think you must also acknowledge that there may be other factors that influence the rate of Axial elongation during OrthoK. Peripheral plus is an important factor but there may be many others that have not been adequately studied.

We acknowledge that. Peripheral Refraction is just a theory. There are many more factors to study in the future. We modified the sentence accordingly.

.. the diameter of the steepened mid-peripheral corneal annulus changed, in which the relative peripheral myopic shift, an increase in HOA, and associated optical factors that had been related to reduce myopia progression in children.

If you have the data, it might be interesting to see if there is any difference in AL elongation between a group of subjects who all have the same pupillary diameter but different PPRD.

Interesting comment. However, it is not easy to find equal values of pupillary diameters, which is why we chose to construct figure 4.

However, we made the requested analysis and sent the image

Jaume Paune

Reviewer 2 Report

This manuscript by Paune et al, reports the effect of different back optical zone diameter of orthokeratology lenses in the control of axial length increase.

There are few issues that need addressing to improve the clarity of the manuscript. 

Major points:

  1. The manuscript looks like there are still some comments from the authors (for example, line 238, Line 255, reference 31). The authors need to go through the manuscript again and address these.
  2. The results section (paragraph 1) states that the samples showed a normal distribution. However, Table 2 presents the results from Wilcoxon test.
  3. It is reported that L-PPRDs were characterized by smaller pupillary diameters than the S-PPRDs. They are independent parameters. How does this affect the results of relative pupil influence? This needs to be discussed further.
  4. Pupil size is obtained from topographic data (through a close view topographer, probability a one-point measurement). It is also reported that the pupil size was smaller. This is discussed as a limitation. Is this way of measuring pupil size enough to extrapolate the findings from this study?

Minor points

  1. First paragraph of introduction: Regarding cause of blindness, it shall be stated as ‘uncorrected myopia’
  2. In the last paragraph in discussion, is it supposed to read choroidal thickness. In that case, the explanation is unclear.

Author Response

We appreciate so much your valuable observations. They help us to a better exposition article. We attach a word document with the observations and changes made on the document. Please see that attachment.

We remain at your disposal for further observations and suggestions

This manuscript by Paune et al, reports the effect of different back optical zone diameter of orthokeratology lenses in the control of axial length increase.

There are few issues that need addressing to improve the clarity of the manuscript. 

In the first place, we appreciated so much all these valuable observations and suggestions. We will try to answer all the questions one by one on the following lines.

Thank you very much for your comment and review.

Major points:

1. The manuscript looks like there are still some comments from the authors (for example, line 238, Line 255, reference 31). The authors need to go through the manuscript again and address these.

Thank you very much for taking care of the review. By mistake you forgot to delete the English proofreader's notes

Comments deleted and revised the entire document

t differences be238 tween the L-BOZD group with a mean myopic increment IN_CREASE? of -0.27±0.23 D and

were ≤4.5 mm. the two groups should not overlap?

  1. Carracedo, G.; Espinosa-Vidal, T.M.; Martínez-Alberquilla, I.; Batres, L. The topograph-ical effect of optical zone diameter in orthokeratology contact lenses in high myopes. J. Ophthalmol. 2019, .ADD THE REST OF THE PUBLISHING DATA

2. The results section (paragraph 1) states that the samples showed a normal distribution. However, Table 2 presents the results from Wilcoxon test.

Sorry for the confusion. It was added that this only happened in the baseline

 “The samples in the baseline showed a normal distribution”

3. It is reported that L-PPRDs were characterized by smaller pupillary diameters than the S-PPRDs. They are independent parameters. How does this affect the results of relative pupil influence? This needs to be discussed further.

Thanks for the comment. In fact it was not described in the methods section. The description of these new parameters was added and described the small and large diameters.

“In order to better perceive the annual increase in the axial length of myopic children, the two analysis parameters (BOZD and PPRD) were divided into two groups (larger and smaller diameters). Thirty-six subjects comprised the group with a BOZD exceeding 5 mm (L-BOZD) and 35 with a BOZD equal to or smaller than 5 mm (S-BOZD). For the plus power ring diameter (PPRD) with larger PPRDs  when they were >4.5 mm (L-PPRD, n=36) and smaller PPRDs when they were ≤4.5 mm (S-PPRD, n=35).”

4. Pupil size is obtained from topographic data (through a close view topographer, probability a one-point measurement). It is also reported that the pupil size was smaller. This is discussed as a limitation. Is this way of measuring pupil size enough to extrapolate the findings from this study?

We appreciate the comment and agree that this is not really the best method to evaluate the pupil diameter. However and considering that this is the method most used in clinical terms in the adaptation of orthokeratology contact lenses by professionals, we understand that the extrapolation of results is the closest to reality. This is clear safeguarding the limitation of the study described.

Thus was added the reference of the description of obtaining the pupil and accrued:

“Pupillary size was obtained directly from the topographic data obtained under photopic conditions (more real conditions for children). (REFERENCIA) There is no ideal method in the literature, the authors consider it to be the most used clinical method in the description of the pupil diameter by professionals.”

Minor points

1. First paragraph of introduction: Regarding cause of blindness, it shall be stated as ‘uncorrected myopia’

Of course it is. It has been corrected to “uncorrected myopia’”

2. In the last paragraph in discussion, is it supposed to read choroidal thickness. In that case, the explanation is unclear.

We agree with the reviewer.

The paragraph was rewritten for better understanding.

Jaume Paune

Reviewer 3 Report

See my document attached.

THis study addresses a very important topic in myopia management. However, there are several major concerns about the methodology and the analysis which must be addressed. Major revision needed.

COnclusions are not supported strongly by the data but with modifications made, as suggested, it may be the case.

Author Response

Response Reviewer 3
THis study addresses a very important topic in myopia management. However, there are several major concerns about the methodology and the analysis which must be addressed. Major revision needed.
COnclusions are not supported strongly by the data but with modifications made, as suggested, it may be the case.

We appreciate the review and comments made.
Some of the suggested changes were made throughout the manuscript, which are in line with the other two reviewers. Your major revision contain some questions that would lead to another article, that was not the intention of the authors and the other reviewers. In regard to this we will review all the information requested, change the criteria and differentiated analyses in a future study as suggested.
We respond your comments in the following document
Thank you very much

Specific comments
Line 47: add preventable before blindness.
Done
Line 48: goal is to prevent abnormal elongation of the eye, not to stop natural growth of the eye

Changed for “identifying a way to prevent abnormal axial elongation of the eye in children is essential”

Line 63-64 There is no evidence that images, all the time, focus in front of the retina. What is right to say is that OK reduce the hyperopic defocus and may then help to modify eye elongation. A -8 D lens with +2 D add will not likely to bring the image in front of the retina.

“, altering the retinal image profile causes off-axis images to reduce the hyperopic defocus and help to modify eye elongation.”

Line 69 PPR is determined arbitrary from the center of the annulus. It would be better to describe it by width and power generated. Authors cite Marcotte-Collard article- they should follow his procedure.
Power profile should be analyzed and max convex power must be reported vs baseline myopia. Ex: baseline -3D max convex +5D
Analysis must be documented by quadrant. We know that the impact is not the same nasal vs temporal and superior vs inferior. Also it is rare to have exactly the same amount of power
generated and the same width of + ring 360 degrees around. It is a mistake to average these values.

In line 69 we foud a diagram showing schematically Plus Power Ring. This is just and schema. Regarding that, in In line 158 we may read: “Tangential topographic maps were retrieved using the Keratron Onda [35] (Keratron, Rome, Italy), by which the points of higher plus power (steeper curvature radius) in the PPR were identified for baseline and 12-month treatment and the PPR diameters (PPRDs) for flat and steep meridians were retrieved and the median value was obtained.”

Considered that baseline was once treatment had stabilization. Line 135 “The baseline parameters were collected once refractive and topographic stabilization was obtained.” what is located in the topography are the point of maximum plus power in the PPR, this is a very exact location compared to the “edge” of this ring. We measured in flat and steep meridian, but not in quadrants because is no sense once what is measured is the “ellipsis” that conforms the PPR. Statistical results was not different using flat or steep measurement. Hence, we utilized only mean value for easy understanding.

The reviewer suggest to follow Marcotte-Collard procedure. These authors had a different objecitve than ours. They would compare optical effects at the corneal level. Thus, they collect topographic tangential maps were collected at baseline and 3 months after fitting. We follow the subjects after fitting stabilization, that is usually between 1 to 3 months, and we are not looking to the topographic change, but the influence of the optics to the abnormal eye grow.

It is very interesting to evaluate the distribution of power change related to axial length grow, this is what had done Wang, J. and co-workers in 2018 and Yang X in 2020 were they not only measured by quadrants but by on 278 sections of the topography.

The question that we addressed here is totally different and is looking for the position of the point of maximum positive power of the PPR, also named in the article the center of the ring because it trends to keep this position.

We acknowledge the interest of the approach that the reviewer suggest and we will work on it in another article.

Line 83 Higher myopes are habitually older… They tend naturally to have reduced growth over time vs those before 10 years old. Also OK is doing great the first year of treatment but tend to stabilize after this. Please consider these elements.

We agree these considerations. In the article cited they compare two age groups: 9 ±1 and 15± 3. Younger subjects progressed faster that older, but inside the same age group higher myops progressed less. We attach a image for clarification.

Althoug, we modify the text in lines 87 for clarification: “Further, previous investigations demonstrating that higher baseline refractive errors display a smaller axial elongation. Younger subjects generally show larger changes of AL compared to older subjects[23].”

Lines 88-90 Please justify more the need for customization: why is so ? To increase ratio 1:1 at 1:2 or 1:3 (not knowing the ideal proportion..) ? TO generate more convex power ? To limit treatment zone, and if so, why ? These elements are logical but are not supported so far in the literature.
Authors must justify more their theoretical basis.

We modify the text for clarification. “...OK lenses with a 6-mm back optical zone diameter (BOZD) trend to follow the ratio 1:1 [24,25], meaning a one diopter change in relative peripheral refraction for one diopter change in PPR and studies evaluating the relative corneal refractive power (RCRP) have pointed to the need for a minimal change of 4.50 D in the RCRP to achieve a chance for 80% myopia control [26]. Hence, customization of OK lenses
to obtain an enhanced peripheral plus zone seems suitable for increasing the treatment efficacy.”

Line 93 It is one thing to influence peripheral refraction and there is another thing to consider OK vs myopia control. We may discuss about the way to evaluate peripheral refraction through a modified corneal profile… which is probably not accurate. There is no study linking modifications of the peripheral refraction in OK with myopia control.

We agree this and understand that a lot of discussion may be done in this sense. We modified the text here:
“spatial distribution of the RCRP rather than the total amount may be more important for stopping myopia progression and that future lenses maybe designed with a smaller central OZ [32] or other specific changes on lens design including the reverse zone. However, the association between AL growth and PPR diameter induced by a customized BOZD has not yet been studied in humans.”.

Line 95 These were commercial regular designs, not customized. So this comparison is probably not so relevant here. Justify more the need for customization in fact.

We answered this in previous questions and text modifications

Line 128 Preferably : this is a retrospective study made on data collected from XX to XX . I doubt that the retrospective study took 4 years!

Yes, this sentence was not well expressed. We changed for:
This is a retrospective study based on data collected from subjects fitted for myopia control with orthokeratology lenses at a Centre Marsden private optometric clinic between March 2012 and October 2016

Line 132 Define how did you determine the lens centration. From the pupil center or from the visual axis ? It must be preferably from the visual axis.. Also what constitutes a “regular” TZ ??

We change the sentence to:
“…centered treatment (<0.5 mm of decentration from visual axis) and a uniform TZ were identified by an expert fitter (JPF).”

Line 133 I don’T understand this limitation to entrance criteria. Partial OK is working even more than regular OK (based on Dr Cho’s studies on higher myopes). Keeping just the successful candidates influence the results. I strongly suggest to revisit this criteria and to include more participants.

We include only subjects that obtain 6/6 vision after OK treatment. Partial correction was out of this study an needs to be adressed in another one. Dr Cho performed this study in higher
myops with partial correction. Here we reach up to -6 D that is inside the normal range for ortho-k

Same for the fact to keep the right eye only. We know that we cannot average both eyes, but it is very interesting to compare the evolution of OD vs OS eye. Both can be slightly different at baseline but to evaluate and to compare their respective evolution over time is precious. I would strongly recommend to include OS and to compare the results with OD.

OD and OS can be compared but this is a subject for other study.

Line 138 It is arbitrary to define 2 groups based on large or smaller BOZD. This does not make sense if we do not consider the pupil at the same time. I would eliminate this approach, and I recommend to keep only the subgroups generated later in the paper: TZ lying within the pupil, TZ aligned with the pupil or being outside. Then 3 groups to be compared.

We prefer to keep this information inside the article instead to remove it. The reason remains on the interest for the scientific comunity about BOZD, changes in cornea and axial lenght grow. This is a hot topic today.
However, we agree totally that the pupil needs to be considered and for this we added the subgroups. Sadly this reduced the sample in each group, meaning the results weak.
We will work in future in this sense.

Line 145 Demographics must be described for each subgroups.

We modified the descriptives accordinlgy. Line 265.

Line 150 Sample size will be determined differently: we have 3 groups and need linear regression with multiple variables.

The sample size for BOZD groups was calculated initially. Once 3 subgroups was made it not reach the minimum required into the group of full influence due the difficulty to keep the PPR inside the pupil and not in the edge. We modified the sentence for clarification:

“To calculate the sample size, we assumed a test power of 0.8, and a significance level of 0.05 (two-tailed). The number of subjects required in each group of BOZD was 34.”

Line 154 and following The basis for this study is to evaluate the difference between BOZD of OK lenses. There is nowhere where we can see how this diameter was determined. It must be explicitly explained. It is not just right to say “according to manufacturer’s recommendations”.
Explain also how the second reservoir may play a role… Centration ? Not certainly for optical reasons… The paper is also customization… We don’t understand how these lenses may differ
from commercial regular Ok lenses. Explain also what is the rationale to use toric peripheral curves, and if and when toric base curve were used. So we need to understand how lenses were designed, what is the fitting approach and what makes these lenses different from others.

We modified the text in this regard:
“Participants were fitted with a DRL (double reservoir lens) design (Precilens, Creteil, France) according to the manufacturer’s protocol that considered the topographic values, refraction, and corneal diameter. These lenses include a second tear reservoir formed after the reverse curve by a flattened curve coupled with a steepened curve. All fittings were optimized until centration and the correct refractive outcomes were achieved. Toric designs with or without toric back optic zone radius were used when necessary to obtain the better treatment as possible DRL lenses are made of a Boston XO (hexafocon A) material with oxygen permeability of 141 barriers, refractive index of 1.415, Rockwell R hardness of 112 units, and wetting angle of 49 degrees measured with the captive bubble method.
DRL design includes customization on BOZD. Reduction of BOZD was obtained increasing the width of peripheral curves to keep overall diameter constant, and no changes in reverse curve width were made. Curvature of the reverse curve was adjusted to keep sagittal depth constant.”

Line 168 : Justify why you used 0.01D threshold instead of 0.25D

The device allows for this more accurate steps. Work in steps of 0.25D means loss of precision.

Line 175 Sepcify if cycloplegic refraction was used only at baseline or also with lenses on, during follow-ups. If so, explain why using cycloplegic OR with lenses (not necessary… )

Any work on refractive outcomes, and specifically in myopia control trends to measure refraction under cycloplegia. Not to do this will result in a flawness of the study.

Line 178-180 I don’t understand this measurement except to evaluate lens flexure ?? Justify

Yes, lens flexure. Text modified for clarification

Line 190 Specify when topography was performed : before or after cycloplegia, before or after immersion ultrasound. Many parameters can be influenced after instillation of drops…

We modified the text for better aclaration. The procedure included topographical captures before cyclopegia and before any other measurement.

Line 195 Pupil diameter measured through topography is not habitually highly accurate. Authors specify that this was done under photopic condition. However, topo maps are better evaluated when measurements are made under mesopic condition. Reconciliate..
We added a extensive rational for this on discussion. This point was common with other reviewers. Just to say, that even topography done in total darkness, the intrinsic light of the topographer makes the measurement a photopic conditions.

Stats Multivariate regression analysis must be performed, considering suggestion to process 3 groups. (in the pupil, aligned or exceeding pupil)

Multiple regression generally explains the relationship between multiple independent or predictor variables and one dependent or criterion variable. A dependent variable is modeled as a function of several independent variables with corresponding coefficients, along with the constant term. Here we have three groups and one dependent variable

In such studies SEM (standard of means) must be used instead of standard deviation…

The two other reviewers said nothing in this regard. We will keep this in SD at this point since there is not significative difference.

Line 213 and following Report demographic data per group, not averaged.

Included now in a new table

Line 230 Table2 Counter intuitive that higher myopic change is associated with larger BOZD…please explain.

This is due to the sample distribution. In any case this helps to reinforce the results of the study. As discussed higher myops trend to increase AL less than lower myops, and group of Large BOZD were with higher myopia than lower. But results show that L_BOZD increased more than S_BOZD. Probably with a reverse situation the results will be with even more difference between groups reinforcing more our results

Based on the values, hard to believe that anterior chamber depth was significantly changed over time. Not habitual in OK lens studies… How was it determined (ultrasound )? This method is highly variable and may induce distortion of the results. OR stats methods are not appropriate.

This is exposed in methods and discussion. We acknoledge that ultrasound is not the best method.

In order to simplify the analysis and to answer to the research question, the only element to consider must be AL evolution over time. This is the only metric that matters. Diopters are accessory. We assume that the lenses were customized.

This is true, and we support the findings on AL. Refraction is a additional parameter.

We will have the amount of + generated in each group – so we can evaluate the dose response. It also proves that the lenses were customized. Let say that you generate more than 1:1 ratio, this means that lenses are customized.

As said before, we included the term customized in methods. Customization was specific for BOZD, no other attemps was made in these sample. Ratio 1:1 change was not the objective of this article.

I would strongly recommend to analyse AL elongation over time, based on where the + is landing (in , aligned, outside the pupil).

This is what we did in three groups. Not ennough participants in one group for a good formal analysis as explained in discussion.

Then we can conclude that customizing helps to get a better control and that smaller treatment zone is to be favored. Also alleviate to use BOZD as a reference …. This is a lens parameter… The important element is the treatment zone generated on the cornea. Like in Figure 3 … redo it using treatment zone size and see what happens. This has to be the element to analyze. Analyze it by quadrant obviously.

We modified the words in conclusion:
“Altering the OK lens design can reliably modify the annular PPRD. The current study provided evidence that a small BOZD reduces the PPRD and improves the effect of OK to slow axial growth in myopia by displacing the steepened annular ring in OK closer to the central zone.”
We appreciate so much your suggestion. At this point and with the comments of the other 2 reviewers we will keep this option for another different article.

Discussion
Lines 312-318…. This is my point. Exactly…. The size (width) and power of this ring makes the difference…. BUT it was not reported that way in the text. So change the analysis but keep this orientation for the discussion,

We enjoy have a point of concordance. Text modified already

Line 323 If this group is match for age and sex you can compare. Otherwise … it is not possible to use it as reference

Good point. We modified the text. “In a previous article and in a sample of Caucasian Spanish children with the same characteristics of age, sex distribution, and refractive at baseline [9],” You may look the characteristics into the article

Line 349 Blink study proved a dose-response in humans as well.
added
Line 362 Authors cannot make this conclusion except if they proceed as suggested (3 groups) and to use treatment zone (and not BOZD) as a reference. Same BOZD may generate different effect on the cornea. So treatment zone – through tangential map analysis- must be the reference point.

This point had been largely answered on previous changes made. The reference of the conclusion is indeed about the PPRD and not BZOD as sugested by the reviewer. Tangential maps was the only utilized.

Round 2

Reviewer 3 Report

This represents an improved version of the previous manuscript. I thank the authors for the modifications made. I have still a few comments that would lead to minor modifications.

Line 86 and elsewhere in the text. Authors talk about HOAs as an entity. It is known that, for myopia control, all HOAs are not similar. More precisely, we have to increase positive spherical aberrations to improve myopia control. I would like to see this specification made throughout the text.

Lines 90-91. Confusing as written. demonstrating that higher baseline refractive errors display a smaller axial elongation... not always true if you have a younger patient (<10 years old) with a high myopia. Progression is fast. Sentence must indicate that older patients tend to be more myopic and are progressing less. It is a mistake to associate high myopia and less axial growth... It all depends on the age of the patient. Please clarify.

Line 100    This is especially true for lower myopes. Specify

102-111    Not only the treatment zone must be customized but also the reservoir volume. It is said in line 131-132 but it comes too late... Reader must understand that it is important to customize the lens design to improve the efficacy of the treatment. this implies many elements such as BOZD but also reservoir, etc.  Clarify

Line 144. Begin a new sentence with It includes..  confusing actually

176.  I really want to see an explanation for this second reservoir. We don't know what it is bringning in the equation. Is there a role for this reservoir ? Increasing pressure in the first reservoir ? Helping only to evaluate lens centration ? It is a unique design and we have to know if this may influence the study results. Would be they the same if the second reservoir was not there?

179. Rephrase... Not good english.

183  Talk about Fatt units vs Barrier

190   Keeping the sag constant... For a single participant or for all participants. If so, what is the value of the sag targetted ?

197. Rephrase... 2 achieve in the same sentence

198  REplace achieve by performed

200  I do not agree with the authors that cyclo is mandatory at each visit. It was done like this and it is ok. But cyclo is mandatory at baseline. Once the final lens is designed, it corrects the refractive error properly, based on cyclo baseline results. If VA with lens on is kept the same, then there is no diopters evolution. If over-correction is needed- with the lens on- that means that diopters increased. There is no need for cyclo then.  Also, cyclo may be routinely repeated for control but we can use tropicamide then instead of cyclopentolate, once baseline results are known. Especially in a cohort of older patients.

Table 1 and analsysis

Interesting to see that larger BOZD is fitted on smaller pupils.... We would guess that it would had been the opposite. This may influence the results and the dose-response of the system. Not sure that this is clearly taken in account in the analysis. Because this means that there is less + reaching the peripheral retina when you have a larger zone fitted on a smaller pupil (Chen study)

To understand more we miss one element. PPRD is well described in mm but we don't know how much + was generated... Would be important, from topography, to describe this value. It may explain in part the results. If the sag was kept constant, we woulde expect that the level of + would be quite similar accross the lens designs.

For PPRD, authors take the average value... Nowhere it is explained that difference between flat and steep PPRD is not significant and then we can average the values... Must be added

Conclusion

I don't really understand the conclusion that authors failed to prove that when PPRD lies in the pupil (Full effect) there is no difference in the outcome... It was quite clear that it was the case... proving the dose effect in humans again. Rephrase to be clearer on this aspect.

Author Response

Dear Reviewer

We attach our response on a word document for your convenience

Best regards

J Paune

Response 2 to Reviewer 3.

This represents an improved version of the previous manuscript. I thank the authors for the modifications made. I have still a few comments that would lead to minor modifications.

We appreciate so much your recognition to our best efforts to improve the article. We are very thankful for your comments that help to make it better.

Line 86 and elsewhere in the text. Authors talk about HOAs as an entity. It is known that, for myopia control, all HOAs are not similar. More precisely, we have to increase positive spherical aberrations to improve myopia control. I would like to see this specification made throughout the text.

We modified this on line 86 and beyond, as in discussion: “ an increase in HOA, more specifically SA, and associated optical factors that had been related to reduce myopia progression in children..”

Even we may agree that Spherical Aberration is probably the most important HOAs related with reduction in axial elongation under ortho-k, it is not clear at 100% which HOA is involved in myopia control. Hiraoka found that Coma had the best correlation, and found a weaker with SA. Meanwhile, Lau (a P Cho collaborator) found SA;  “… the potential role of HOA, particularly spherical aberration, as the possible mechanism of slowing axial elongation in ortho-k treatment.”. We should need more studies to understand about.

Lines 90-91. Confusing as written. demonstrating that higher baseline refractive errors display a smaller axial elongation... not always true if you have a younger patient (<10 years old) with a high myopia. Progression is fast. Sentence must indicate that older patients tend to be more myopic and are progressing less. It is a mistake to associate high myopia and less axial growth... It all depends on the age of the patient. Please clarify.

We change the sentence for: “higher baseline refractive errors treated with orthokeratology trends for a smaller axial elongation. Younger subjects generally show lower myopia and larger changes in AL compared to older subjects[23].”

This is again a controversial subject. Indeed younger trend to increase myopia faster and trend to have lower refractive error. The reference; Queirós, A.; et al. Refractive, biometric and corneal topographic parameter changes during 12 months of orthokeratology. Clin. Exp. Optom. 2020, 103, 454–462. would be interesting to have look at, because it shows that for same group of age, higher baseline errors experience less AL elongation compared to lower myopia at baseline.

Moreover, in the latest article of P. Cho (2020) she shows the difference between groups of fast, medium or slow progressors. Fast progressors had the better result on efficacy terms, and this group had younger subjects, but the 3 groups has similar refractive error. Thus, for same refractive error, younger trends to progress faster and had better results on retardation for AE.

Here the point is: At same age, higher refractive error (probably meaning faster progressors?) slow down better. ¿why?

Line 100    This is especially true for lower myopes. Specify

Done

102-111    Not only the treatment zone must be customized but also the reservoir volume. It is said in line 131-132 but it comes too late... Reader must understand that it is important to customize the lens design to improve the efficacy of the treatment. this implies many elements such as BOZD but also reservoir, etc.  Clarify

We included the differences in lens designs as sigmoidal corneal proximity “return zone,” or reverse curves.

We specify that in this case, the lens utilized changes BOZD, but not reverse curve width and a very small adjustment on radius of curvature for reverse curve. In other words, we agree that a full customization is necessary to increase + power, but in this article we doesn’t change this.

Line 144. Begin a new sentence with It includes..  confusing actually

Modified

  1. I really want to see an explanation for this second reservoir. We don't know what it is bringning in the equation. Is there a role for this reservoir ? Increasing pressure in the first reservoir ? Helping only to evaluate lens centration ? It is a unique design and we have to know if this may influence the study results. Would be they the same if the second reservoir was not there?

We changed the sentence to:

Participants were fitted with a DRL (double reservoir lens) design (Precilens, Creteil, France) according to the manufacturer’s protocol that considered the topographic values, refraction, and corneal diameter. These lenses include a second tear reservoir formed after the reverse curve by a flattened curve coupled with a steepened curve. Accordingly with the manufacturer, second tear reservoir increase hydrodynamic suction forces, what improves centration and faster epithelial changes. All fittings were optimized until centration and the correct refractive outcomes were achieved. Toric designs with or without toric back optic zone radius were used when necessary to obtain the better treatment as possible DRL lenses are made of a Boston XO (hexafocon A) material with oxygen permeability of 100  ISO/Fatt , refractive index of 1.415, Rockwell R hardness of 112 units, and wetting angle of 49 degrees measured with the captive bubble method.

DRL design allows customization of BOZD. Reduction of BOZD is obtained increasing the width of peripheral curves to maintain total diameter constant, and no changes in reverse curve width were made. Curvature of the reverse curve was adjusted to remain relationship with a given cornea unchanged.

  1. Rephrase... Not good english.

OK

183  Talk about Fatt units vs Barrier.

Yes, this was wrong. Also the Dk, it should be 100

190   Keeping the sag constant... For a single participant or for all participants. If so, what is the value of the sag targetted ?

Modified. This was keep the lens with same relationship with the cornea.

  1. Rephrase... 2 achieve in the same sentence

Changed

198  Replace achieve by performed

Changed

200  I do not agree with the authors that cyclo is mandatory at each visit. It was done like this and it is ok. But cyclo is mandatory at baseline. Once the final lens is designed, it corrects the refractive error properly, based on cyclo baseline results. If VA with lens on is kept the same, then there is no diopters evolution. If over-correction is needed- with the lens on- that means that diopters increased. There is no need for cyclo then.  Also, cyclo may be routinely repeated for control but we can use tropicamide then instead of cyclopentolate, once baseline results are known. Especially in a cohort of older patients.

This is clearly a choice. Tropicamide is easier patients because is shorter effect and is ok for studies. Cycloplegia is necessary if a monitorization of the refraction is done. we considered it is an older cohort of subjects, but they had 13 years. We would be sure that they aren’t not accommodating. In fact, we doesn’t measure the vision without lenses. We measured the residual refraction (over refraction) with lenses on. Due to compression factor, many of the times the value was positive.

Table 1 and analysis

Interesting to see that larger BOZD is fitted on smaller pupils.... We would guess that it would had been the opposite. This may influence the results and the dose-response of the system. Not sure that this is clearly taken in account in the analysis. Because this means that there is less + reaching the peripheral retina when you have a larger zone fitted on a smaller pupil (Chen study)

In a prospective design we will do the opposite. But, sadly, the data is what it is. In that time we fitted the lenses with larger diameter in all the subjects.

And this is why it is reduced the number of subjects on group with full PPR pupil influence. Of course it influence the results for worse. It means that the results are more prone to be right. In a different sample probably it will have been more strong results.

To understand more we miss one element. PPRD is well described in mm but we don't know how much + was generated... Would be important, from topography, to describe this value. It may explain in part the results. If the sag was kept constant, we would expect that the level of + would be quite similar across the lens designs.

Yes, here + power was the regular result of a orthok lens. No differences on + power when the reduction of diameter was done. Besides the peripheral refraction profile was changed due the position, but not the quantity of change. Sadly we didn’t measure this time. The issue for scientific studies is that one thing is changed at time. Here we changed the diameter of PPR. In other studies they looked to changes in + in periphery and no change in position for PPRD

For PPRD, authors take the average value... Nowhere it is explained that difference between flat and steep PPRD is not significant and then we can average the values... Must be added

We came back to our data and re-done the analysis. Once we did the statistics for flat (horizontal) and steep (vertical) meridian we found that horizontal PPRD was statistically significant regarding AL, vertical wasn’t. Mean was close. P=0.053. Nevertheless, in regard to refractive outcome, it was amazingly statistically significant with p<0.001 in all the groups showing a dose-response of the system.

We modified this part of the article including the graph for horizontal PPR

We would give the thanks to this reviewer to insist in this point.

Conclusion

I don't really understand the conclusion that authors failed to prove that when PPRD lies in the pupil (Full effect) there is no difference in the outcome... It was quite clear that it was the case... proving the dose effect in humans again. Rephrase to be clearer on this aspect.

We modified the text accordingly to the news findings. Now it is clear a dose effect and the influence of the pupil on the AL and refractive evolution.

Thanks again
